# The Gambia has eliminated trachoma as a public health problem: Challenges and successes

**Agatha Aboe[1], Balla Musa Joof [2], Sarjo Kebba Kanyi[3], Abba Hydara [3], Philip Downs [4]\*, Simon Bush[1], Paul Courtright[5]**

**1** Sightsavers, Accra, Ghana, **2** Sightsavers, Banjul, The Gambia, **3** National Eye Health Programme, Ministry of Health, Banjul, The Gambia; Sheikh Zayed Regional Eye Care Centre, Kanifing, The Gambia, **4** Sightsavers, Durham, North Carolina, United States of America, **5** Kilimanjaro Centre for Community Ophthalmology, Division of Ophthalmology, University of Cape Town, Cape Town, South Africa

\* pdowns@sightsavers.org

**Data Availability Statement:** All historical trachoma treatment and survey data, as reported to WHO, are publicly accessible via the elimination of neglected tropical diseases (ESPEN) website

## Abstract

Trachoma is the leading infectious cause of blindness in the world and has been known to be a major public health problem in The Gambia for over 60 years. Nationwide blindness surveys, including trachoma, in 1986 and 1996 provided the foundation for a comprehensive plan to implement a trachoma elimination strategy. Impact and pre-validation surveillance surveys in 2011–13 demonstrated that active trachoma was below WHO threshold for elimination but trichiasis remained a public health problem. Trichiasis-only surveys in 2019 demonstrated that trichiasis was below WHO thresholds for elimination and in 2020 the Government of The Gambia completed and submitted its dossier for validation of elimination as a public health problem. Challenges that The Gambia faced on the pathway to elimination included effective use of data for decision making, poor trichiasis surgical outcomes, lack of access to antibiotic treatment for low prevalence districts, high attrition of ophthalmic nurses trained as trichiasis surgeons, unexpected active trachoma in madrassas, the misalignment of elimination of active trachoma and trichiasis, trichiasis in urban settings, and maintaining the quality of surgery post-elimination when trichiasis cases are rare. Elimination of trachoma does not end with the submission of an elimination dossier; The Gambia will need to sustain monitoring and support over the coming years.

## Author summary

The World Health Organization has validated that trachoma is no longer a public health problem in The Gambia as of April 2021. In this article, the authors summarize critical milestones achieved by The Gambia in its journey to elimination and the challenges to implementing the WHO-endorsed elimination strategy, including surgical management for people with trichiasis, antibiotics to all endemic communities, and uptake of face-washing practices and environmental improvements. In 1986, a national survey of blindness and low vision in The Gambia found that 17 per cent of blindness was caused by

(https://espen.afro.who.int/countries/the-gambia).
Data inquiries can be submitted to ESPEN Team
Lead Dr. Amir Bedri Kello at kelloa@who.int. All
other data and related metadata underlying
reported findings are provided as part of the
submitted article.

**Funding:** The author(s) received no specific
funding for this work.

**Competing interests:** The authors have declared
that no competing interests exist.

trachoma in the form of corneal opacity due to trichiasis. At this time, eye health services delivered by a trained eye health were limited to the Greater Banjul area, meaning people in remote locations were frequently left behind. This changed with the coordination of partners and donors strengthening the knowledge base of health workers on trachoma preventative measures and the capacity of ophthalmic nurses to deliver trichiasis surgical support to all trachoma-endemic communities. Final epidemiological surveys conducted in 2019 confirmed that there was sufficient evidence to show that elimination thresholds had been achieved all endemic areas prompting the formal submission of the country's elimination dossier to the WHO.

## Introduction

Trachoma is the leading infectious cause of blindness in the world. [1] The causative agent is *Chlamydia trachomatis*, with infection most common in young children; infection can lead to trachomatous inflammation—follicular (TF), the clinical sign used for estimating the presence of active trachoma. After repeated infections, the upper eyelid tarsal conjunctiva develops scars which, over time, may cause the upper eyelid to become deformed. Deformed upper eye-lids can result in trichiasis, the presence of eyelashes touching the front of the eye. Two decades ago, it was estimated that globally 8 million people had trichiasis and were at risk of blindness from corneal ulceration and scarring. [2].

The WHO Alliance for the Global Elimination of Trachoma by 2020 (GET2020), established in 1996 and endorsed in 1998 by the World Health Assembly [3], called for the elimination of trachoma as a public health problem by 2020 and endorsed a four-pronged strategy referred to as the SAFE strategy. SAFE is an acronym which includes Surgery (S), to correct deformed lids; Antibiotics (A) to treat active trachoma and reduce the community burden of infection; Facial (F) cleanliness and Environmental (E) improvement to reduce transmission within communities.

Elimination of trachoma as a public health problem requires clear, evidence-based documentation, showing through survey that (a) TF prevalence in children aged 1–9 years is <5% in all nationally defined administrative units that are suspected to have trachoma after having stopped MDA for at least 2 years, (b) trichiasis prevalence unknown to the health system in adults aged 15 years and above is less than 0.2% in all previously trachoma endemic administrative units, and (c) providing evidence of the ability to manage incident trichiasis cases emerging in communities [4].

The Gambia is a small West African country, spread out east to west along the Gambia River. Its current population is estimated at 2.5 million, with about 31,000 living in the urbanized areas of Banjul, the capital. [5] For the purpose of health-related interventions there are 7 health regions which comprise 42 health districts.

## Methods

Trachoma had been recognized as a major public health problem in The Gambia for a long time; a survey in 1959 in a rural village, Marakissa showed that the prevalence of active trachoma among children aged 0–9 years was 65.7%. [6] Trachoma research was carried out in The Gambia through the Medical Research Council facility in Fajara. These efforts led to trachoma activities being integrated in the work-plans for the National Eye Health Programme. As the programme evolved, there was increased collaboration with the other NTDs

programmes, the Directorate of Health Promotion and Education, Primary Health Care Unit and the Epidemiological and Disease Control Programme of Ministry of Health.

The Gambia was one of the first countries in sub-Saharan Africa to undertake a nationwide blindness survey (1986); the age-sex adjusted prevalence of blindness (<3/60 in the better eye) was 0.7%. [7] In this survey, trachoma, in the form of corneal opacity due to trichiasis, was responsible for 17% of blindness. A second nationwide blindness survey (1996) showed a slight reduction in overall prevalence of blindness (0.55%) with trachoma accounting for 5% of blindness. [8] The pooled (country-wide) prevalence of active trachoma in children aged 0–14 years reduced from 10.7% in 1986 to 4.8% in 1996.

The findings from the 1996 nationwide blindness survey, in addition to a targeted trachoma survey in 2006 in two health regions [9] formed the foundation for a five-year plan of action to control trachoma in The Gambia. The surveys suggested that 5 of the 7 regions had active trachoma, requiring intervention. The 2006 survey did not measure trichiasis, but the 1996 nationwide survey indicated that all regions had trichiasis, requiring intervention.

We describe the trachoma elimination experience in The Gambia, with a particular focus on key challenges that the programme faced on its pathway to elimination.

## Results

In June 1986, following the first national survey of blindness and eye diseases, the Ministry of Health established the National Eye Care Programme (NECP) in The Gambia, now renamed the National Eye Health Programme (NEHP). The NEHP has responsibility, among other activities, for the overall coordination and management of trachoma elimination activities across the country. Like the global effort at trachoma elimination through the SAFE strategy, the initial steps toward trachoma control and elimination started in The Gambia in 1997, with the NECP conducting baseline mapping as part of the implementation of the Global Elimination of Trachoma by the year 2020 (GET2020) strategy. The successful completion of the baseline mapping exercise gave the NECP the strongest evidence on *where* and *who* had trachoma. Even though interventions still mainly focused on facial cleanliness and environmental improvement, some community ophthalmic nurses began offering lid surgeries using the standardized Trabut technique for the surgical management of trachomatous trichiasis. The national trachoma elimination plan (January 2007 to December 2009) was developed to guide the programme towards achieving trachoma elimination by the year 2020. The government sought to ensure there were adequately trained human resources at various levels of the primary health care system for the implementation of all components of the SAFE strategy. The government also provided some of the needed infrastructure, vehicles, and other logistics for programme implementation as well as the monthly remuneration of health workers and other staff. With support from partners, government provided free antibiotic treatments during mass drug administration (MDA) of azithromycin for trachoma. Trichiasis surgery was provided free of charge. All survey protocols, the collection and reporting of findings from surveys and supervision and monitoring of trachoma field activities were carried out by government workers with technical support from partners.

Partners such as Sightsavers, the International Trachoma Initiative (ITI), the Medical Research Council, and the London School of Hygiene and Tropical Medicine assisted the programme to draft a strategy for the elimination of trachoma, support for activities related to the SAFE strategy, provision of donated antibiotics, and research. The Gambia's success in eliminating trachoma is largely attributable to strong partnerships that the Ministry of Health established for implementation of the SAFE strategy. Additional partnerships included UNICEF and WHO and government departments such as the Department of

Community Development under the Ministry of Local Government and Lands; the Department of Water Resources; the Ministry of Basic and Secondary Education; and the National Environment Agency. At the heart of the implementation of the SAFE strategy were the Gambian community volunteers, referred to as *nyateros* (from the Mandinka meaning "Friend of the Eye"), who played a crucial role in mobilizing communities and promoting behavior change via traditional communication, music and drama (strategies of behavioural change communication).

## Surgery

Based on baseline survey data, it was estimated that there were over 1,000 people with trichiasis throughout the country, although the wide confidence interval around the prevalence figure suggests that number of trichiasis cases to be imprecise. Trichiasis was a public health problem in all seven health regions surveyed in 1996.

In The Gambia, as in other trachoma endemic countries, trichiasis surgery is usually performed by non-physician trained clinical officers or ophthalmic nurses. The initial training on trichiasis surgery was included in the curriculum within The Gambia's Regional Ophthalmic Training Program, ensuring integration, standardization, and sustainability. There was practical training to reinforce surgical skills and trainees were required to perform a minimum of 20 surgeries under supervision. From 2001 to 2020, 94 Gambian ophthalmic nurses were trained as trichiasis surgeons. Field research demonstrated that the surgical quality of many surgeons was sub-optimal, with post-operative trichiasis detected in 46% of eyes assessed. [10] Based on these results a consultant ophthalmologist re-trained selected surgeons.

Trichiasis case finding was conducted by a network of trained community-based workers (CBWs) including schoolteachers, *nyateros*, Village Health Workers and Integrated Eye Health Workers under the supervision of ophthalmic nurses and others. The CBWs were given two days of training in identification of trichiasis cases. After training, they were provided with visual acuity charts, torchlights, and spare batteries as well as SAFE strategy and WHO Primary Eye Care posters. In many settings, case finding was done via a door-to-door visit, with those suspected of having trichiasis given referrals to trichiasis surgeons for confirmation and surgery as indicated. Persons with trichiasis who presented at health facilities were also managed. All trichiasis cases and management provided were documented in trichiasis registers. This is part of the key performance indicators reported by the NECP.

During the programme implementation, all trichiasis surgeons received three instrument sets plus consumables, which were replaced periodically, as indicated. Throughout the country, trichiasis surgery was provided on an outreach basis and at specific healthcare facilities. The policy of the NEHP was to operate on all eyes with five or more eyelashes touching the eyeball, one or more eyelashes touching the cornea, a history of epilation, or where there was post-operative trichiasis. For individuals with one eye suitable for surgery under this policy, both eyes could be operated at the same time. All operated cases received a single oral dose of azithromycin when donated antibiotic was available in the country. Epilation was offered to persons with trichiasis who did not otherwise meet the criteria for an offer of surgery. Patients with serious medical conditions such as diabetes or hypertension were advised to wait until their conditions were controlled before surgery would be recommended. It was further recommended that complicated cases, including patients with postoperative trichiasis, should be managed by ophthalmologists.

To maintain quality of care, the programme instituted a quarterly supervision system for trichiasis surgeons. Starting in 2009 routine patient follow up, at one day, 7–14 days, and 3–6 months was initiated as a programme activity. Surgeon audits were not implemented.

The Gambia provided between 152 and 569 surgeries each year over the eight years of the trichiasis programme (2009–2016). From 2017 trichiasis surgeries continued to be provided during routine eye care outreaches or at designated health facilities, with diminishing numbers reported. Surveys conducted in 2013 showed that five of the nine health zones still had not met the WHO criteria for elimination of trichiasis as a public health problem, leading to more systematic active case finding and surgery.

## Antibiotics

Starting in 2007, Pfizer-donated azithromycin (Zithromax) was supplied through the International Trachoma Initiative (ITI) for MDA. MDAs took place over a four-year period from 2007–2010, totaling 370,675 doses. Treatment was done annually for one year or three years, according to WHO recommendations: tablets or pediatric oral suspension for all adults and children 6 months of age or older. Children under 6 months of age and pregnant women were supplied 1% tetracycline eye ointment, applied twice daily for 6 weeks. Decisions on which districts to treat were based on the findings from the 1996 and 2006 surveys. In 11 districts with a prevalence of TF of ≥10% in children aged 1–9 years of age, three annual MDAs were undertaken. In thirteen districts with a prevalence of TF of 5–9.9% in children aged 1–9 years, one MDA round was undertaken. In these thirteen districts each community underwent screening for TF and if the prevalence in children was ≥10%, the entire community was treated. When the prevalence was less than 10%, only households with one or more TF cases were treated. This atypical approach to MDA was part of research into different methods for MDA. [11]

Prior to any MDA event, communities were sensitized using messages delivered via community radio, traditional communicators, *nyateros*, religious leaders, and VHWs. Posters, billboards, and flip charts were developed and distributed prior to the MDA to communicate to the community the importance of MDA. On the day of MDA, traditional communicators and *nyateros* moved from house to house encouraging community members to participate in the MDA. Drug distributors were provided a two-day training consisting of classroom training as well as field training for hands-on demonstration and practice. All drug distributors, volunteers and supervisors were provided with an allowance as a form of motivation. Drug distribution was conducted using a combination of the 'pulled crowd' approach at various centres within the community, as well as door-to-door for mop-up treatments to ensure high coverage. National, regional and district level staff of the Ministry of Health and their partners provided supervision. Over the four-year period no severe adverse events were reported. Standardized MDA reporting forms were distributed to each drug distribution team. The forms helped to collect data on households visited, number of persons treated, and doses given. All data were reported to the district health management team, for the regional and the national level. That said, data on antibiotic coverage was not maintained by the programme.

Impact surveys, conducted in 2011 and 2012, showed the prevalence of TF was less than 5% in all previously endemic health zones and MDA was discontinued.

## Facial cleanliness & environmental improvements

In 1999 a national multi-sector taskforce was established for planning, coordination and implementation of community engagement and social mobilization activities for trachoma prevention and control. The key messages centered on: daily face washing with clean water and soap, proper hand washing at critical times, keeping the environment clean, keeping domestic animals away from houses or human dwellings, provision for proper location and use of a pit latrine, and good health seeking behaviour. A broad mix of communication channels and methods were used, including posters, leaflets and radio messages. These channels

were used extensively to raise awareness and provide information to promote personnel hygiene, especially for face washing, facial cleanliness, and handwashing. CHWs and peer health educators were actively engaged in promoting face washing at community level and in schools. For effective communication, local languages were used by traditional communicators, such as *nyateros*. The network of health workers, traditional communicators and peer health educators were instrumental in promoting behavioural change.

The Government of The Gambia recognized that the availability of safe water, improved sanitation and hygiene were essential in achieving improvements in public health and socio-economic development and it prioritized access to water and sanitation through national development blueprints. It supported the concept: "access to safe water and basic sanitation is critical for basic survival" and, in collaboration with national and international partners, expanded coverage of water supply and sanitation. This included providing communities with access to potable water through the construction of boreholes, hand pumps, and solar water pumping systems. In 2018, national access to improved water supply stood at 90.4%, of which coverage was 92.2% in urban areas and 86.9% in rural areas. The national coverage for improved sanitation was 61.8%, with 73.9% in urban areas and 36.6% in rural areas. [12] In June 2019 the Government of The Gambia, with partner support, initiated a $ 1.8 billion clean water and sanitation project to improve the socio-economic and environmental conditions of the rural and peri-urban population by improving access to sustainable water and sanitation infrastructures and services.

## Reaching elimination

Elimination of trachoma as a public health problem is clearly defined by WHO, including both active disease (TF) as well as trichiasis. Confirming that the elimination thresholds have been achieved requires evidence both that the prevalence of TF in 1-9-year-olds is below 5%, and that the prevalence of trichiasis unknown to the health system in $\geq$15-year-olds is below 0.2%. (4) For TF, an impact survey of an evaluation unit showing a TF prevalence <5% must be followed, after at least two years, by a pre-validation surveillance survey confirming that TF has remained below the elimination threshold. Impact surveys are scheduled according to the timing of the final MDA. In 2011, 2012 and 2013 surveys were undertaken to assess the status of trachoma in the country and findings were used to demonstrate that elimination of TF had been achieved after categorizing the surveys as impact and pre-validation surveillance surveys.

Of the nine evaluation units surveyed between 2011 and 2013 (Fig 1), five had not attained the WHO trichiasis prevalence threshold for elimination as a public health problem. In specific areas of these districts considered to be "high risk" for trichiasis, the programme conducted community-based case finding followed by provision of surgery or other management. Trichiasis-only surveys [13] conducted in 2018–2019, ascertained that the WHO threshold for elimination of trachoma had been fully met throughout the country. (Table 1 and Fig 1).

Notably, Banjul, Kanifing and other urban areas were excluded from the evaluation units for the 2011–2013 surveys although these areas were included in the blindness surveys conducted in 1986 and 1996. Earlier surveys had shown the urban areas to have trichiasis of public health significance and surgical interventions were initiated to manage any remaining cases.

In 2020 the Ministry of Health and its partners completed the WHO trachoma elimination dossier and submitted it to the relevant WHO offices for assessment. Currently, The Gambia joins Ghana as two sub-Saharan African countries that have been validated for elimination of trachoma as a public health problem. [1] Post validation surveillance in The Gambia is being supported by the Ministry of Health through integrated data management systems, building capacity amongst health workers, strengthening cross-border collaboration and conducting

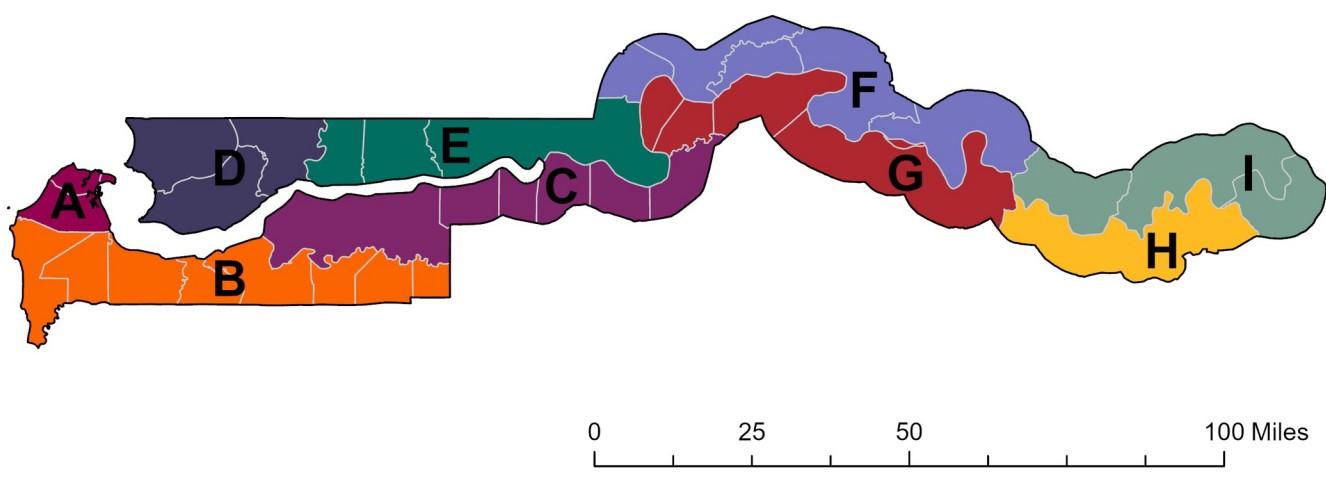

**Fig 1. Map of The Gambia showing survey areas in 2013.** Base layer: https://geodata.ucdavis.edu/gadm/gadm4.0/shp/gadm40_GMB_shp.zip.

supportive supervision. Forms have been developed for reporting on TF, to be used by eye nurses only, and for reporting on trichiasis, by both eye and non-eye workers. Approximately 160 eye health and other workers have been trained on surveillance.

At national level, indicators for trachoma (TF and trichiasis) have been integrated into the DHIS2 system since 2017. Trachoma is a notifiable disease with all health regions reporting on a weekly basis, supported by a WhatsApp surveillance group. Private health care providers are encouraged to report any suspected cases of TF or trichiasis. All suspected cases of TF and trichiasis are to be seen, reviewed, and confirmed by senior eye care staff before the data is entered into the national DHIS2 platform.

Trichiasis surgeries continue to be provided in all the previously endemic regions, which is evidence that the health system can identify and manage incident trichiasis cases. As of early 2021 there were 46 eye care workers prepared to perform trichiasis surgery on walk-in patients.

Preliminary results from the recently concluded Gambia National Eye Health Survey 2019 (3rd National Eye Survey) did not indicate any findings that suggest re-emergence of TF or clustering of TT throughout the country.

**Table 1. Findings from 2013 pre-validation surveillance surveys and 2019 trichiasis-only surveys (in zones with a trichiasis prevalence ≥0.2% in adults aged 15 years or more in 2013).**

| Zone | 2013 TF prevalence % (95% CI) | 2013 trichiasis prevalence % (95%CI) | 2019 trichiasis prevalence % (95%CI) | 2019 trichiasis prevalence (unknown to the health system) % (95% CI) |
|---|---|---|---|---|
| A | 1.2 (0.7–2.2) | 0.3 (0.1–0.8) | 0.05 (0.00–0.16) | 0 |
| B | 1.9 (1.2–3.0) | 0.6 (0.3–1.1) | 0.08 (0.01–0.15) | 0.02 (0.00–0.05) |
| C | 1.8 (1.1–3.1) | 1.0 (0.6–1.9) | 0.18 (0.08–0.31) | 0 |
| D | 3.2 (2.3–4.4) | 0.1 (0.04–0.5) | | |
| E | 0.2 (0.1–0.9) | 0.3 (0.1–0.8) | 0.12 (0.02–0.26) | 0 |
| F | 0.6 (0.3–1.3) | 0.4 (0.2–0.9) | 0.18 (0.06–0.28) | 0.02 (0.00–0.04) |
| G | 2.1 (1.4–3.1) | 0.1 (0.03–0.4) | | |
| H | 0.5 (0.2–1.0) | 0 | | |
| I | 0.4 (0.2–0.8) | 0.1 (0.04–0.3) | | |

## Discussion

The pathway to the present involved considerable challenges, many of which other countries will also face. Each challenge required government officials, partners, surgeons, trainers, and others to consider how best to address the problem given limited financial and human resources.

### Challenge #1: Effective use of data for decision making

With the generation of data on trachoma throughout the country, The Gambia was ready to intervene but, in the late 1990s there was little global guidance on criteria for where and how to intervene. Global alliances and the introduction of initiatives to address trachoma were drafted in the late 1990s but there was insufficient detail for effective and efficient intervention. In the face of this challenge The Gambia decided to initiate activities related to facial cleanliness and environmental improvement, a good start to trachoma elimination.

### Challenge #2: Poor trichiasis surgical outcomes

The finding of poor surgical outcomes [10] threatened the success of the trachoma control programme. The unexpected finding of a large proportion of surgical cases having poor outcomes, in particular for specific surgeons, meant that the surgical programme had to be redesigned in order to ensure that good quality became a focus within the program. Poor quality trichiasis surgery threatens surgical uptake throughout trachoma-endemic countries and review of outcomes through surgeon audits [14] is critical for success. It also requires quarterly nationwide monitoring and evaluation visits by supervising ophthalmologist. Defining key technical and administrative roles and encouraging joint integrated supervisory visits (at central and regional levels) would ensure early identification of challenges and instituting remedial measures.

### Challenge #3: Lack of access to antibiotic treatment for low prevalence districts

The Gambia sought to manage districts with a TF prevalence of 5–9.9% in children aged 1–9 years, but the request was before the ITI made Zithromax available for routine treatment in districts with TF 5–9.9%. With support from the research community, a screen-and-treat approach was adopted. The screen-and-treat approach required considerable personnel with sufficient clinical skills to examine children in all communities, then decide whether the entire community needed treatment or not. The excess manpower and research structure was expensive to implement.

### Challenge #4: High attrition of ophthalmic nurses (previously trained as trichiasis surgeons)

After scaling up the training and placement of community ophthalmic nurses, high attrition occurred. The government addressed the challenge by providing incentives to the ophthalmic nurses, especially those posted to rural communities. In health facilities without ophthalmic nurses, general nurses were trained as integrated eye health workers. Attrition of trained eye care workers remains a problem in many trachoma endemic countries and innovative approaches are needed to address the challenge. The Gambia MoH under various directors, prioritized midwifery services over ophthalmic services and this was repeatedly manifested in the delayed or lack of promotion for trained ophthalmic nurses. As a consequence, many trained ophthalmic nurses had to go back to do midwifery courses in order to get a promotion.

Thus, many found their renewed paths in midwifery-related activities against their primary career wishes.

### Challenge #5: Unexpected active trachoma in madrassas

The focus of the school-based trachoma control activities was on formal, public schools but it was recognized that cases of active trachoma were reported from madrassas, especially those with boarding facilities. Based on this finding, madrassa boarding facilities were included in subsequent community screening and treatment activities.

### Challenge #6: Misalignment of elimination of active trachoma and trichiasis

The WHO requirement that all previously endemic countries must fulfill two prevalence-based criteria to declare elimination of trachoma: [a] prevalence of TF in children aged 1 to 9 years <5%, and [b] prevalence of trachomatous trichiasis unknown to the health system in adults aged 15 years and above <0.2% challenged The Gambia programme as these two indicators could not be fulfilled at the same time. Throughout the country, the WHO threshold for active trachoma was achieved first, after which population-based surveys were discontinued. This limited the ability of the programme to demonstrate trichiasis elimination. Full geographic coverage [4] of trichiasis management (all cases found and managed) could not be demonstrated as only limited case finding was undertaken. Consequently, trichiasis-only surveys were used to demonstrate elimination. Many trachoma endemic countries face a similar challenge brought on by the differential speeds at which populations reach the elimination prevalence thresholds for TF and trichiasis.

### Challenge #7: Trichiasis in urban settings

The national eye surveys in 1986 and 1996, covered urban areas as well as rural areas. Trachoma surveys, generally posit that urban areas are not trachoma-suspect and are not mapped for trachoma. The early eye surveys indicated that the trichiasis prevalence in Banjul was higher than the WHO threshold for elimination. Relying on old survey data challenged decision-making. A number of trachoma endemic countries were in a similar situation and were unsure if trichiasis is below the elimination threshold in urban centres or not. It is most probable that early surveys conducted in urban areas like Banjul overestimated the prevalence of TT by virtue of their methodology and analysis plan at the time.

### Challenge #8: Maintaining the quality of surgery post-elimination when trichiasis cases are rare

Like all previously endemic countries, The Gambia needs to sustain trichiasis services post-elimination but it can be quite difficult to maintain the quality of surgery when surgeons have few cases to manage. It is likely that, as incident cases continue to decrease, the proportion of cases that are postoperative trichiasis (i.e. had previous surgery) will increase. These cases require more specialist interventions by ophthalmologists, currently stationed at the Sheikh Zayed Regional Eye Care Centre in Kanifing. Monitoring incident and complicated cases will inform decisions by the eye care programme managers on how to meet these needs effectively and efficiently. Again, the emphasis on a joint monitoring and evaluation team composed of ophthalmologists and programme administrators would go a long way in dealing with these problems.

## Conclusion

The government of The Gambia, working together with other governmental and non-governmental partners have, over the past 20 years implemented the SAFE strategy and attained elimination of trachoma as a public health problem. The challenges encountered along this pathway are not unique to The Gambia. Like all trachoma endemic countries, the complexity of the interventions required coordination of manpower and resources to implement the SAFE strategy and to conduct the impact and pre-validation surveillance surveys required to demonstrate that elimination had been achieved. The official designation by WHO is that trachoma elimination has been achieved, yet The Gambia shall continue to monitor progress.

## Acknowledgments

We would like to thank the Ministry of Health, The Gambia and the health workers and community volunteers who worked so hard to provide the data and information that have been used in this paper. Particular thanks to Hannah Faal, Victoria Goode, the late Bakary Cham, Ansumana Sillah, Abdoulie Dibba, Awa Nije, Momodou Bah, and the late Sheriff Trawally for their invaluable work that led to the elimination of trachoma in The Gambia.

## Author Contributions

**Conceptualization:** Agatha Aboe, Philip Downs, Paul Courtright.

**Data curation:** Balla Musa Joof, Paul Courtright.

**Formal analysis:** Sarjo Kebba Kanyi, Abba Hydara.

**Investigation:** Sarjo Kebba Kanyi, Abba Hydara.

**Project administration:** Sarjo Kebba Kanyi, Abba Hydara.

**Resources:** Simon Bush.

**Supervision:** Abba Hydara.

**Writing – original draft:** Agatha Aboe, Paul Courtright.

**Writing – review & editing:** Agatha Aboe, Balla Musa Joof, Sarjo Kebba Kanyi, Abba Hydara, Philip Downs, Simon Bush, Paul Courtright.

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
