## [Decision Letter · Decision Letter 0]

31 Oct 2021

Dear Dr. Downs,

Thank you very much for submitting your manuscript "The Gambia has eliminated trachoma as a public health problem: Challenges and successes" for consideration at PLOS Neglected Tropical Diseases. As with all papers reviewed by the journal, your manuscript was reviewed by members of the editorial board and by several independent reviewers. The reviewers appreciated the attention to an important topic. Based on the reviews, we are likely to accept this manuscript for publication, providing that you modify the manuscript according to the review recommendations. 

Please revise in line with comments made by reviewer 1

Sincerely,

Robin L. Bailey

Associate Editor

Michael Marks

Deputy Editor

Please revise in line with comments made by reviewer 1

Reviewer's Responses to Questions

**Key Review Criteria Required for Acceptance?**

**Methods**

-Are the objectives of the study clearly articulated with a clear testable hypothesis stated?

-Is the study design appropriate to address the stated objectives?

-Is the population clearly described and appropriate for the hypothesis being tested?

-Is the sample size sufficient to ensure adequate power to address the hypothesis being tested?

-Were correct statistical analysis used to support conclusions?

-Are there concerns about ethical or regulatory requirements being met?

Reviewer #1: (No Response)

Reviewer #2: The objectives are clearly stated and the other criteria do not apply to a review of this kind

Reviewer #3: The paper gives background history of trachoma control activities in the Gambia and implementation of SAFE strategy for over 20 years. The authors describe the challenges faced by the trachoma programme in reaching the elimination thresholds - especially of TT as well as how the programme tried to address the challenges faced. It highlights the importance of collaboration of different stakeholders in the implementation of the SAFE strategy.

**Results**

-Does the analysis presented match the analysis plan?

-Are the results clearly and completely presented?

-Are the figures (Tables, Images) of sufficient quality for clarity?

Reviewer #1: (No Response)

Reviewer #2: No data analysis plan is needed for a review such as this

Reviewer #3: The authors have clearly presented the historical background of trachoma control in the Gambia as well as listed the challenges faced by the country and how they addressed those in its effort towards being validated for having eliminated trachoma as a public health problem in April 2021.

**Conclusions**

-Are the conclusions supported by the data presented?

-Are the limitations of analysis clearly described?

-Do the authors discuss how these data can be helpful to advance our understanding of the topic under study?

-Is public health relevance addressed?

Reviewer #1: (No Response)

Reviewer #2: The conclusions are clearly presented and the manner in which the Gambian eye care programme overcame the many challenges identified to achieve the elimination of trachoma as a public health problem will be extremely helpful to other national programmes

Reviewer #3: The conclusions presented are supported by issues discussed in the paper. The authors discuss how their findings could be of help to other trachoma endemic countries that might face similar challenges in their effort to eliminate trachoma as a public health problem.

**Editorial and Data Presentation Modifications?**

Reviewer #1: Thank you for the opportunity to review this paper. I have only minor suggestions

Line 37: suggest remove the parentheses

Line 47: please correct “Chlamydia trachomatous” to “Chlamydia trachomatis”

Line 48: the hyphen in “trachomatous inflammation-follicular” should be corrected to and type-set as an em-dash

Line 49: suggest change “the clinical sign used to denote the presence of active trachoma” to “the clinical sign used for estimating the prevalence of active trachoma”

Line 65: delete “old” (since you have “aged” in front of “1-9 years”)

Lines 64-67: criterion (a) and criterion (b) should each include the word “prevalence”

Line 68: please delete “or other”, to conform with WHO guidance on the elimination criteria

Line 107: please change “completion of the GET2020 exercise” to “completion of the baseline mapping exercise”

Line 110: please change “TRABUT” to “Trabut” 

Line 119: since azithromycin is a generic name, please use a lower case “a”. 

Line 144: suggest change “integrated” to “included”, since integration is used subsequently in the same sentence to identify one of the positive outcomes of this inclusion

Line 151, suggest remove the inverted commas that currently surround the word “surgeons” to avoid the possibility that this is read with a sarcastic tone

Line 171: suggest edit “Epilation was offered to persons with fewer than five eyelashes touching the

eyeball if no eyelashes touched the cornea” to read “Epilation was offered to persons with trichiasis who did not otherwise meet the criteria for an offer of surgery”

Line 250: suggest edit, “Confirming that the elimination thresholds have been achieved requires survey of both children aged 1-9 years old to demonstrate that the prevalence of TF is below 5%, and survey of adults aged 15 years and above to demonstrate that the prevalence of trichiasis unknown to the health system is below 0.2%.” to read “Confirming that the elimination thresholds have been achieved requires evidence both that the prevalence of TF in 1-9-year-olds is below 5%, and that the prevalence of trichiasis unknown to the health system in ≥15-year-olds is below 0.2%.” (The point being that surveys aren’t necessarily mandatory if other high-quality evidence can be furnished.)

Line 255: please change “after two years” to “after at least two years”

Line 353: please edit “The WHO requirement that all previously endemic countries must fulfill two criteria” to read “The WHO requirement that all previously endemic countries must fulfill two prevalence-based criteria…” (The point here is that there are three criteria for elimination, the third being the existence of a system to detect and manage incident cases of TT. But the two identified in this sentence are the two prevalence-based criteria.)

Line 363: suggest change “misalignment of survey results for TF and trichiasis” to “differential speeds at which populations reach the elimination prevalence thresholds for TF and trichiasis”. (It is not the fault of the surveys.)

Line 364: I would be tempted to delete challenge 7. The issue in Banjul was almost certainly that the early surveys, by virtue of their methodology and analysis plan, overestimated the prevalence of TT

Line 382: please delete “of the” or change “comprising of the” to “composed of”

Reviewer #2: There is a typo in line 47, this should be Chlamydia trachomatis not trachomatous. The final sentence in the Conclusion section seems to be missing a word.

Reviewer #3: In order to avoid misunderstanding of the criteria for elimination of trachoma as a public health problem, the authors are advised to revise the sentences on Page 3, Lines 64-66: it should contain the fact that the prevalence of TF in 1-9 years old children of <5% in previously endemic districts after having stopped MDA for 2 years. 

Page 3 , Line 68: better to specify that it is "trachomatous trichiasis" and not "other trichiasis cases"

Page 11, Line 283: elimination of trachoma as a public health problem

Page 13, Line 352: trachomatous trichiasis 

Page 15, Line 392: elimination of trachoma as a public health problem

**Summary and General Comments**

Reviewer #1: (No Response)

Reviewer #2: This paper clearly describes the challenges faced by the Gambian national eye care programme in achieving the elimination of trachoma as a public health problem, and the way in which each of these challenges was overcome. This paper will be of great value to eye care programmes facing similar challenges in other trachoma endemic countries as they attempt to achieve the elimination of trachoma by 2030, as proposed in the WHO NTD roadmap

Reviewer #3: This article comes at a time when more and more trachoma endemic countries are reaching the last mile towards achieving elimination of trachoma as a public health problem. The challenges faced by the Gambia and the solutions implemented could be helpful for other countries that have similar challenges.

PLOS authors have the option to publish the peer review history of their article (what does this mean?). If published, this will include your full peer review and any attached files.

Reviewer #1: No

Reviewer #2: Yes: David Mabey

Reviewer #3: Yes: Amir B Kello

Figure Files:

Data Requirements:

Reproducibility:

References

---

## [Editor Report · Decision Letter 1]

26 Feb 2022

Dear Dr. Downs,

We are pleased to inform you that your manuscript 'The Gambia has eliminated trachoma as a public health problem: Challenges and successes' has been provisionally accepted for publication in PLOS Neglected Tropical Diseases.

Best regards,

Michael Marks

Deputy Editor

Michael Marks

Deputy Editor

---

## [Editor Report · Acceptance letter]

24 Mar 2022

Dear Dr. Downs,

We are delighted to inform you that your manuscript, "The Gambia has eliminated trachoma as a public health problem: Challenges and successes," has been formally accepted for publication in PLOS Neglected Tropical Diseases.

Best regards,

Shaden Kamhawi

co-Editor-in-Chief

Paul Brindley

co-Editor-in-Chief
